**Data Availability Statement:** All data and supporting information files are available from the Open Science Framework database (https://osf.io/mdwfc/).

# Determinants of community members' willingness to donate stool for faecal microbiota transplantation

**Melissa K. Hyde**[1]*, **Barbara M. Masser**[1,2]

1 School of Psychology, The University of Queensland, Brisbane, QLD, Australia, 2 Clinical Services and Research, Australian Red Cross Lifeblood, Sydney, NSW, Australia

* m.hyde@uq.edu.au

## Abstract

Universal stool banks rely on, but face difficulties recruiting, community volunteers to donate stool for faecal microbiota transplantation (FMT) to effectively treat recurrent Clostridioides difficile. This study sought to identify determinants of community members' willingness to donate stool to guide donor recruitment. 397 Australian residents (52% male, 47% 21–30 years, 63% university educated) completed a survey to gauge willingness to donate stool, bowel habits, information needs, attitudes, barriers, and motives for donation. Most reported regular bowel movements (BMs; 90%), morning BMs (63%), BMs ≤5 minutes duration (67%), and some discomfort doing BMs in public restrooms (69%). Less than half were willing to donate stool in-centre (45% willing) or at home (48%). Important information needs identified by >80% were convenience and travel requirements associated with donation. Main barriers were logistics, capabilities to donate, disgust (e.g., donation process), and discomfort (e.g., privacy). The main motivator was altruism, with compensation secondary. Linear regression models identified less discomfort doing BMs in public restrooms (β = -0.15), understanding benefits to patients (β = 0.15), placing less importance on understanding the donation process (β = -0.13), and positive attitudes (β = 0.56) as determinants of willingness to donate in-centre. Understanding benefits to self (β = 0.11) and patients (β = 0.24), placing less importance on understanding the donation purpose (β = -0.19), and positive attitudes (β = 0.50) determined willingness to donate at home. Stool banks should consider donor's bowel habits, comfort donating in-centre, and information needs early in recruitment; and implement flexible logistics for potential donors who face time constraints and limited access to stool banks.

## Introduction

Faecal microbiota transplantation (FMT) is a highly effective treatment for the 20%-30% of patients who develop recurrent Clostridioides difficile infection (CDI) after first-line treatment with antibiotics [1], and shows promise for other inflammatory bowel diseases and gastrointestinal conditions [2–4]. Successful treatment of recurrent CDI and prevention of recurrence occurs for 70–90% of FMT recipients [1], as well as health care savings following FMT (e.g.,

**Funding:** The authors received funding from The University of Queensland Vice-Chancellor's Strategic Funds (BM) and internal funds granted by The University of Queensland School of Psychology (MH) to collect data using a paid participant panel and to employ a research assistant to assist with data collection and coding. The funders had no role in study design, data collection and analysis, decision to publish, or preparation of the manuscript. There was no additional internal or external funding received for this study.

**Competing interests:** The authors have declared that no competing interests exist.

42% reduction in hospital costs in the first year) [5], and substantially improved quality of life [6]. Widespread availability of this increasingly in-demand treatment depends on the availability of willing stool donors. Traditionally, stool has been donated by someone known to the patient. More recently, volunteer donors from universal stool banks are the preferred and more advantageous approach due to stringent screening and fewer delays in access to treatment [1,7]. However, recruitment of volunteer stool donors is challenging and costly [8], numbers of eligible donors rarely meet demand [1], and little information is available about what factors influence people's decisions to donate stool.

Helping patients and extending current blood donor activities to try stool donation are identified key motivators of stool donors [9], although other research shows that stool donor's altruistic behaviour (volunteering, donating blood) was not associated with donation frequency or quantity of donations processed as treatments [10]. Medical students [11], and university-affiliated health professionals, students and staff [12], who had greater awareness of FMT or its benefit for patients were more willing to donate stool (58% highly willing to donate [12]). Willingness was predicted by male gender, altruism, preference for economic compensation, positive attitudes towards FMT, blood donation, having considered organ donation, and less concern about barriers to donation (stool collection as unpleasant, invasive screening process, monthly donation as a large commitment) [12]. One-third identified logistics such as the required time commitment and delivery of their donation, as the main barriers to stool donation [12]. Logistics, including the required donation frequency and duration, also prevent potential community donors from progressing beyond pre-screening [8,13–16].

Individual bowel habits such as the frequency, timing, and duration of bowel movements (BM), as well as discomfort donating outside the home environment (e.g., at a collection facility), likely also impact logistics and donation willingness, but their impact has not been considered. Moreover, there is variability in stool collection procedures with some facilities allowing donors to collect stool at home and transport it to a facility [17,18], while others adhere to biosafety regulations that require stool collection to occur in a designated room at the facility [19]. As such, it is important to consider both home and in-centre donation. Regarding bowel habits, general population studies suggest that most people fall within the range of 3 BMs per day to 3 per week, and have morning BMs [20–22], although reports vary depending on measures used [20,23,24], and gender or age differences [21–23,25]. Patients who failed to give a sample for stool banking either could not do a BM, or did a BM outside the designated collection period [26]. Further, consumer surveys report the general public's discomfort doing a BM away from home due to lack of privacy or embarrassment, with males more comfortable using a public restroom for BMs than females [27].

Given the increasing reliance on community donors to support universal stool banks and FMT programs, the aims of this study were to provide guidance for donor recruitment by examining willingness to donate stool in-centre and at home, and identifying potential contributors to this willingness, particularly bowel habits, information needs, motives and barriers to donation. In doing so, this study also provides the first insight into the Australian community's willingness to donate stool.

## Materials and methods

Human Research Ethics Approval was obtained from The University of Queensland (Approval number 2020000029) and Australian Red Cross Lifeblood (Approval number 2020#03) committees. Participants indicated their consent to participate by selecting a response confirming their consent prior to commencing a survey and by submission of a survey.

This is a cross-sectional study using data obtained from Australian residents aged ≥18 years who completed an online survey (Qualtrics platform). Residents were recruited via the online crowdsourcing platform, Prolific, which connects researchers with community members who have registered to participate in research and receive monetary compensation for their time. Prolific verifies and checks the quality of data provided by participants to ensure that it is high quality. Australian residents registered on Prolific (approximately 1200 active in the past 90 days; 48% male) initially received a notification about the study "*Understanding community attitudes towards stool donation*", and were invited to complete a short pre-screening survey to determine if they were eligible to participate. Eligibility criteria were as follows: currently living in Australia; aged 18 years or older; consider yourself to be in good health with no current medical conditions; consider yourself to be of normal weight; are not currently taking medication for a medical condition; do not currently have a chronic digestive disorder or condition that may affect how often you do a bowel movement; and eligible to donate blood.

While the eligibility criteria employed in this study broadly reflected screening criteria for recruitment of community donors to stool banks, it should be noted that there is variability in screening criteria globally. For instance, the Australian Consensus Working Group [28] and the Standard for Faecal Microbiota Transplant Products (TGO 105) [29] states that generally donors should be aged between 16–60 years and donors over the age of 50 should have completed bowel cancer screening. The international consensus conference on stool banking for FMT [17] states younger individuals aged below 50 years (or below 60 years if bowel cancer screening has been completed) are preferred as potential donors. The FMT-standardization Study Group's Nanjing consensus on methodology of washed microbiota transplantation [19] recommends healthy adults and adolescents, preferably those aged 6–24 years, as FMT donors.

Once participants self-selected into the study and confirmed their eligibility (S1 Appendix), those eligible provided consent to participate by selecting the yes response to confirm that they had read a participant information sheet, understood the topic of the research, understood that their data would be reported in a de-identified form, and agree to participate. Upon consenting, residents were then invited to complete the main survey. Surveys were completed from February 17th to March 18th, 2020, and participants were compensated £6.00 per hour for their time. University human research ethics committee approval was obtained prior to study commencement. Data from a sub-group of these participants that examines a different set of variables regarding the role of ambivalence in eligible blood donors' decisions to donate stool has been published elsewhere [30].

Survey questions were drawn from published studies on stool donation or from other contexts and adapted. Participant characteristics were assessed using one-item measures of age (in years), gender, education, BMI (kg/m$^2$), current blood donor and registered organ donor status. Self-reports of bowel habits comprised six items that measured frequency of BMs [27,31,32]: whether BMs were regular and occurred every day (yes, no, sometimes), and on average, the daily and weekly frequency of BMs (free-text); and two items assessed when in the day participants usually or most often had a BM (morning, midday, afternoon, evening, night) [27], and approximate duration of BMs from start to finish (minutes). A 4-item sub-scale [33] assessed discomfort having a BM in a public restroom (e.g., I cannot use the toilet in a public restroom to have a BM when other people are around), 0 none of the time to 4 all of the time, Cronbach's α = 0.91.

Awareness of CDI and FMT prior to the study was established using two items, scored yes, no, unsure [32]. Following this, all participants read short paragraphs describing CDI [34–36], FMT [37–39], and typical requirements for donors and donation to ensure equivalency of knowledge before indicating willingness to donate (S2 Appendix). Willingness to donate to a stool bank was measured via one-item scales across scenarios that varied location: at a

collection facility (in-centre) (unknown recipient), at home and deliver to a facility (unknown recipient); and purpose: for a loved one, research, or to develop new treatment(s). Participants indicated their willingness on a slider bar from 0 to 100, where a score of 0 indicated participants were not at all willing to donate, and a score of 100 indicated participants were extremely willing to donate (S1 Appendix). Those indicating willingness to donate stool in-centre or at home (i.e., a score on the scale mid-point of 50 or above) were then presented with questions that asked them to report using free-text responses how often and for how long they would be willing to donate in each scenario. Participants were given an example of how to format their response to ensure clarity and consistency of responses across participants (S1 Appendix). For example, a person willing to give 2 times a day for 1 week would write "2 times a day" in the first free-text box and "1 week" in the second free-text box. One author (MKH) manually coded the free-text responses separately for frequency (e.g., daily and 1 time a day were coded as "1 time a day") and length of time (e.g., 2 weeks and a fortnight were coded as "2 weeks"). These codes were then manually combined (e.g., "1 time a day for 2 weeks"). Given there was great variability in responses and some were unclear, responses were further coded as: a set number of times, daily, weekly, monthly, yearly, other, or unclear.

Following indicating their willingness to donate, participants rated the importance of nine information items when considering stool donation (e.g., understanding how donating stool could help patients), 1 not at all important to 7 very important [40]. Participants self-reported their general attitudes towards stool donation (no opinion to very positive) and their attitudes towards personally donating (negative, neutral, positive) on 1-item scales [41]. Motives and barriers to donation were identified using two free-text response questions "if you were willing to donate to a stool bank for a person you did not know who was sick with CDI, what would be your main reason(s) for doing so?" and "what would stop you or make it difficult to donate?".

### Statistical analysis

Categorical (frequencies, percentages) and continuous (means, standard deviations) variables were summarised using descriptive statistics. Cronbach's alpha $\geq 0.70$ confirmed scale reliability. Differences in responses by category were explored using chi-square tests of independence with follow-up z-tests, and continuous variables were examined using univariate analysis of variance with post-hoc Bonferroni tests. Linear regression analysis identified determinants of willingness, with $p \leq 0.05$ indicating significance. Exploratory analyses identified potential differences in responses by gender (male, female), age group ($<21$, 21–30, 31–40, 41–50, $>50$) [12], university education (yes, no), discomfort doing BMs in public restrooms (none, some/a little, most/all, of the time), blood and organ donor status (yes, no/unsure).

Free-text qualitative responses were analysed using thematic analysis [42]. Two coders independently generated themes, and these were compared and refined, with disagreements resolved by a third coder. Themes described by $\geq 10\%$ of participants are reported.

### Results

A total of 758 Australian residents received survey invitations, of which 491 met study criteria (64.7% eligible). Of those eligible, 397 provided complete data for analysis (81% response rate). Participants mean age was 29.59 years (SD = 9.33, 18–73). Mean BMI was 24.04 (SD = 3.86), and 65% were in the 'healthy' BMI range (18.5 to 24.9).

Almost 90% reported regular BMs. Over half had one BM per day and one-third had seven BMs per week, most had morning BMs, of $\leq 5$ minutes duration. On average, participants experienced some discomfort having BMs in public restrooms (M = 2.71, SD = 1.23, range

1–5, higher scores indicating more discomfort); 69% experienced discomfort at least a little of the time. Table 1 shows participant's bowel habits overall and by gender and age group. More females (vs. males) had morning BMs and took ≤5 minutes doing so. Fewer participants <21 years had morning BMs, and more participants in this age group had BMs in the evening and at night (vs. other age groups). Significantly more females (vs. males) experienced discomfort most to all of the time having BMs in public restrooms.

Participants appeared neither willing nor unwilling to donate stool in-centre at a stool bank or at home and delivering it, and mean willingness scores did not differ significantly (Table 2). Of those scoring ≥50 on willingness to donate in-centre ($n = 212$), 61% would donate weekly (24% daily, 7% monthly, 8% a set number of times). Participants willing to donate weekly ($n = 129$), would donate once (60%) or twice weekly (27%); and 73% would donate for 4 to 12 weeks. Similarly, of participants scoring ≥50 on willingness to donate at home ($n = 215$), 64% would donate weekly (31% daily, 4% monthly, 1% a set number of times). Those willing to donate weekly ($n = 137$) would do so once (36%), twice (31%), or three times (16%) weekly; and 72% would donate for 4 to 12 weeks. Table 2 shows participant characteristics by willingness to donate stool when location (in-centre, at home) and purpose for donating vary (loved one, research, new treatments). Willingness differed by blood and organ donor status, bowel habits, discomfort using public restrooms, awareness of CDI and FMT.

Information needs rated as most important (≥6 on the 7-point scale) were convenience of donating (81%) and needing to travel to donate (80%). One-third had very positive attitudes towards donating stool in general and felt positively about personally donating. Table 3 displays participant's information needs and attitudes towards donating stool. Information needs differed by gender, age group, and education, whereas general attitudes varied by gender and age, and personal attitudes by age.

Table 4 presents hierarchical linear regressions of predictors of willingness to donate stool for a stool bank in-centre and at home. Participant characteristics, bowel habits, and knowledge of FMT were entered into the model at step 1, followed by information needs and attitude to personally donating stool in step 2. At the final model step, significant predictors of willingness to donate in-centre were less discomfort doing a BM in a public restroom, understanding how donating stool could help patients, less need to understand the donation process, and positive attitude towards personally donating ($R^2 = 0.45$). For willingness to donate at home, understanding the personal benefits of donation and for patients, less need to understand the aim of donating stool, and positive attitudes towards personally donating were significant predictors at the final model step ($R^2 = 0.37$). Participant characteristics were significant predictors initially but did not remain so once information needs and attitude were added.

Responses indicating the main reason(s) participants would consider donating stool were thematically analysed and two of seven motives identified were described by >10% of participants. The first main theme, altruism, reflected predominantly impure altruism with participant's desire to help others generally, or specifically contribute to the recovery of those sick. The second main theme, compensation, focused predominantly on payment as either a primary or secondary motivator in addition to helping others. Table 5 details themes identified for motives and exemplar quotes.

Responses detailing factors that would make it difficult or stop participants from donating were thematically analysed and four of six barriers identified were described by >10% of participants (Table 5). The main barrier, logistics, captured inconvenience generally and specifically related to travel for donation purposes, and lack of flexibility in scheduling/donation requirements. The second barrier, concerns about capabilities, reflected participant's apprehension that the process would be too difficult or complicated, or they would be unable to successfully complete it. The third and fourth barriers, disgust and discomfort, comprised

**Table 1. Participant's frequency, duration and timing of bowel movements by gender and age (N = 397).**

| Bowel habits | All n (%) | Gender | | Age (years) | | | | |
|---|---|---|---|---|---|---|---|---|
| | | Male n (%) | Female n (%) | <21 n (%) | 21–30 n (%) | 31–40 n (%) | 41–50 n (%) | >50 n (%) |
| **Regular BM** | | | | | | | | |
| Yes | 355 (89.4) | 188 (90.4) | 167 (88.4) | 55 (88.7) | 165 (89.2) | 88 (89.8) | 35 (89.7) | 12 (92.3) |
| No | 10 (2.5) | 7 (3.4) | 3 (1.6) | 1 (1.6) | 7 (3.8) | 1 (1.0) | 1 (2.6) | 0 (0.0) |
| Sometimes | 32 (8.1) | 13 (6.3) | 19 (10.1) | 6 (9.7) | 13 (7.0) | 9 (9.2) | 3 (7.7) | 1 (7.7) |
| *Between sub-group comparison* | | *p* = 0.216 | | *p* = 0.921 | | | | |
| **Daily frequency of BMs** | | | | | | | | |
| <1 | 20 (5.0) | 10 (4.8) | 10 (5.3) | 2 (3.2) | 12 (6.5) | 5 (5.1) | 1 (2.6) | 0 (0.0) |
| 1 only | 231 (58.2) | 118 (56.7) | 113 (59.8) | 33 (53.2) | 107 (57.8) | 52 (53.1) | 28 (71.8) | 11 (84.6) |
| 1–2 | 14 (3.5) | 5 (2.4) | 9 (4.8) | 1 (1.6) | 8 (4.3) | 3 (3.1) | 2 (5.1) | 0 (0.0) |
| 2 only | 96 (24.2) | 53 (25.5) | 43 (22.8) | 16 (25.8) | 44 (23.8) | 29 (29.6) | 5 (12.8) | 2 (15.4) |
| >2 | 36 (9.1) | 22 (10.6) | 14 (7.4) | 10 (16.1) | 14 (7.6) | 9 (9.2) | 3 (7.7) | 0 (0.0) |
| *Between sub-group comparison* | | *p* = 0.530 | | *p* = 0.419 | | | | |
| **Weekly frequency of BMs** | | | | | | | | |
| <7 | 95 (23.9) | 41 (19.7) | 54 (28.6) | 12 (19.4) | 50 (27.0) | 23 (23.5) | 8 (20.5) | 2 (15.4) |
| 7 only | 112 (28.2) | 59 (28.4) | 53 (28.0) | 14 (22.6) | 47 (25.4) | 26 (26.5) | 20 (51.3) | 5 (38.5) |
| 7–14 | 83 (20.9) | 48 (23.1) | 35 (18.5) | 15 (24.2) | 42 (22.7) | 16 (16.3) | 6 (15.4) | 4 (30.8) |
| 14 only | 56 (14.1) | 30 (14.4) | 26 (13.8) | 10 (16.1) | 25 (13.5) | 18 (18.4) | 1 (2.6) | 2 (15.4) |
| >14 | 51 (12.8) | 30 (14.4) | 21 (11.1) | 11 (17.7) | 21 (11.4) | 15 (15.3) | 4 (10.3) | 0 (0.0) |
| *Between sub-group comparison* | | *p* = 0.276 | | *p* = 0.119 | | | | |
| **Timing of usual BM** | | | | | | | | |
| In the morning | 251 (63.2) | 121 (58.2) | 130 (68.8)[a] | 24 (38.7)[d] | 119 (64.3) | 70 (71.4) | 25 (64.1) | 13 (100) |
| Midday | 29 (7.3) | 20 (9.6) | 9 (4.8) | 4 (6.5) | 18 (9.7) | 5 (5.1) | 2 (5.1) | 0 (0.0) |
| In the afternoon | 53 (13.4) | 33 (15.9) | 20 (10.6) | 10 (16.1) | 24 (13.0) | 13 (13.3) | 6 (15.4) | 0 (0.0) |
| In the evening | 50 (12.6) | 30 (14.4) | 20 (10.6) | 16 (25.8)[e] | 21 (11.4) | 8 (8.2) | 5 (12.8) | 0 (0.0) |
| At night | 14 (3.5) | 4 (1.9) | 10 (5.3) | 8 (12.9)[f] | 3 (1.6) | 2 (2.0) | 1 (2.6) | 0 (0.0) |
| *Between sub-group comparison* | | **p = 0.023** | | **p < 0.001** | | | | |
| **Timing of additional BM (if more than one) (n = 258)** | | | | | | | | |
| In the morning | 48 (18.6) | 29 (19.6) | 19 (17.3) | 10 (22.7) | 15 (12.5) | 16 (24.2) | 7 (31.8) | 0 (0.0) |
| Midday | 27 (10.5) | 13 (8.8) | 14 (12.7) | 7 (15.9) | 14 (11.7) | 4 (6.1) | 1 (4.5) | 1 (16.7) |
| In the afternoon | 74 (28.7) | 41 (27.7) | 33 (30.0) | 8 (18.2) | 39 (32.5) | 18 (27.3) | 7 (31.8) | 2 (33.3) |
| In the evening | 71 (27.5) | 40 (27.7) | 31 (28.2) | 11 (25.0) | 28 (23.3) | 24 (36.4) | 6 (27.3) | 2 (33.3) |
| At night | 38 (14.7) | 25 (16.9) | 13 (11.8) | 8 (18.2) | 24 (20.0) | 4 (6.1) | 1 (4.5) | 1 (16.7) |
| *Between sub-group comparison* | | *p* = 0.668 | | *p* = 0.100 | | | | |
| **Duration of BM (minutes) (n = 394)** | | | | | | | | |
| ≤ 5 | 264 (67.0) | 117 (56.8) | 147 (78.2)[b] | 37 (60.7) | 114 (62.0) | 71 (72.4) | 30 (78.9) | 12 (92.3) |
| 6–10 | 96 (24.2) | 63 (30.6) | 33 (17.6) | 17 (27.9) | 48 (26.1) | 22 (22.4) | 8 (21.1) | 1 (7.7) |
| >10 | 34 (8.6) | 26 (12.6) | 8 (4.3) | 7 (11.5) | 22 (12.0) | 5 (5.1) | 0 (0.0) | 0 (0.0) |
| *Between sub-group comparison* | | **p < 0.001** | | *p* = 0.067 | | | | |
| **Discomfort doing a BM in a public restroom** | | | | | | | | |
| Discomfort none of the time (1) | 125 (31.5) | 72 (34.6) | 53 (28.0) | 14 (22.6) | 56 (30.3) | 36 (36.7) | 13 (33.3) | 6 (46.2) |
| Discomfort a little/some of the time (2) | 190 (47.9) | 105 (50.5) | 85 (45.0) | 28 (45.2) | 92 (49.7) | 47 (48.0) | 16 (41.0) | 7 (53.8) |
| Discomfort most/all of the time (3) | 82 (20.7) | 31 (14.9) | 51 (27.0)[c] | 20 (32.3) | 37 (20.0) | 15 (15.3) | 10 (25.6) | 0 (0.0) |
| *Between sub-group comparison* | | **p = 0.011** | | *p* = 0.121 | | | | |

BM = Bowel movement. Bolded numbers indicate a statistically significant difference.

[a] female vs. male, z = 2.2, *p* = 0.0287.

[b] female vs. male, z = 4.5, *p* < 0.001.

[c] female vs. male, z = 3.0, *p* = 0.003.

[d] < 21 years vs. 21–30 (z = 3.5, *p* <0.001), 31–40 (z = 4.1, *p* <0.001), 41–50 (z = 2.5, *p* = 0.013), or 50 (z = 4.0, *p* <0.001) years.

[e] < 21 years vs. 21–30 (z = 2.7, *p* = 0.006), 31–40 (z = 3.0, *p* = 0.002), or 50 (z = 2.1, *p* = 0.04) years.

[f] < 21 years vs. 21–30 (z = 3.7, *p* < 0.001), or 31–40 (z = 2.8, *p* = 0.005) years.

**Table 2. Willingness to donate stool (0 not at all willing to 100 extremely willing) when location (in-centre to a stool bank, at home and deliver to a facility) and purpose (for a loved one, research, develop new treatment/s) are varied by participant characteristics (N = 397).**

| Characteristic | Number of participants (%) | Location | | Purpose | | |
|---|---|---|---|---|---|---|
| | | In-centre (recipient unknown) | At home & deliver (recipient unknown) | Loved one* | Research | New treatments |
| **All participants** | | 52.43 (30.60) | 52.05 (31.83) | 85.19 (22.74) | 59.18 (31.04) | 61.97 (31.01) |
| At all willing (≥60 on scale) | | 180 (45%) | 190 (48%) | 354 (89%) | 218 (55%) | 238 (60%) |
| Highly willing (≥80 on scale) | | 105 (26%) | 111 (28%) | 306 (77%) | 139 (35%) | 159 (40%) |
| **Gender** | | | | | | |
| Male | 208 (52.4) | 53.04 (30.11) | 50.07 (32.38) | 85.39 (22.94) | 57.93 (30.77) | 62.27 (30.33) |
| Female | 189 (47.6) | 51.76 (31.20) | 54.25 (31.16) | 84.98 (22.57) | 60.55 (31.37) | 61.63 (31.83) |
| *Between sub-group comparison* | | *p* = 0.677 | *p* = 0.192 | *p* = 0.858 | *p* = 0.402 | *p* = 0.839 |
| **Age (years)** | | | | | | |
| <21 | 62 (15.6) | 45.14 (28.46) | 42.45 (28.86) | 80.63 (26.50) | 49.02 (31.24) | 53.39 (30.90) |
| 21–30 | 185 (46.6) | 53.32 (29.46) | 54.15 (31.43) | 85.22 (21.08) | 59.83 (29.28) | 63.17 (29.25) |
| 31–40 | 98 (24.7) | 54.58 (32.71) | 52.31 (31.87) | 88.47 (19.51) | 63.01 (31.54) | 65.38 (31.39) |
| 41–50 | 39 (9.8) | 52.79 (31.05) | 55.26 (32.94) | 83.87 (28.14) | 61.67 (32.94) | 60.87 (34.69) |
| >50 | 13 (3.3) | 57.08 (37.93) | 56.62 (42.50) | 85.92 (29.77) | 62.00 (39.05) | 63.38 (38.60) |
| *Between sub-group comparison* | | *p* = 0.348 | *p* = 0.130 | *p* = 0.323 | *p* = 0.72 | *p* = 0.179 |
| **University education** | | | | | | |
| No | 149 (37.5) | 49.40 (29.15) | 48.24 (30.92) | 84.30 (23.74) | 55.82 (31.91) | 59.45 (31.43) |
| Yes | 248 (62.5) | 54.25 (31.36) | 54.35 (32.21) | 85.73 (22.14) | 61.20 (30.39) | 63.48 (30.72) |
| *Between sub-group comparison* | | *p* = 0.127 | *p* = 0.064 | *p* = 0.542 | *p* = 0.095 | *p* = 0.210 |
| **Donated blood in the past year** | | | | | | |
| No/Unsure | 346 (87.2) | 50.52 (30.81) | 51.08 (32.36) | 84.80 (23.07) | 57.03 (31.40) | 60.15 (31.47) |
| Yes | 51 (12.8) | 65.33 (25.95) | 58.63 (27.40) | 87.86 (20.31) | 73.75 (24.09) | 74.27 (24.68) |
| *Between sub-group comparison* | | **p = 0.001** | *p* = 0.114 | *p* = 0.370 | **p < 0.001** | **p = 0.002** |
| **Registered organ donor** | | | | | | |
| No/Unsure | 271 (68.3) | 48.30 (30.04) | 48.26 (31.81) | 83.87 (23.49) | 54.19 (31.25) | 57.42 (31.50) |
| Yes | 126 (31.7) | 61.32 (30.03) | 60.18 (30.44) | 88.03 (20.83) | 69.90 (27.82) | 71.75 (27.62) |
| *Between sub-group comparison* | | **p < 0.001** | **p < 0.001** | *p* = 0.090 | **p < 0.001** | **p < 0.001** |
| **Aware of CDI** | | | | | | |
| No/Unsure | 337 (84.9) | 50.72 (30.54) | 50.43 (32.09) | 85.14 (22.62) | 57.87 (30.99) | 60.69 (31.13) |
| Yes | 60 (15.1) | 62.00 (29.40) | 61.15 (28.90) | 85.50 (23.56) | 66.52 (30.55) | 69.15 (29.58) |
| *Between sub-group comparison* | | **p = 0.008** | **p = 0.016** | *p* = .910 | **p = .047** | *p* = .051 |
| **Aware of FMT** | | | | | | |

*(Continued)*

**Table 2.** (Continued)

| Characteristic | Number of participants (%) | Willingness to donate stool Mean (SD) | | | | |
| | | Location | | Purpose | | |
| | | In-centre (recipient unknown) | At home & deliver (recipient unknown) | Loved one* | Research | New treatments |
|---|---|---|---|---|---|---|
| No/Unsure | 205 (51.6) | 46.21 (28.95) | 46.60 (32.16) | 83.01 (24.75) | 52.55 (30.52) | 55.66 (31.42) |
| Yes | 192 (48.4) | 59.06 (31.00) | 57.85 (30.51) | 87.53 (20.17) | 66.26 (30.10) | 68.70 (29.19) |
| *Between sub-group comparison* | | ***p* < 0.001** | ***p* < 0.001** | *p* = .048 | ***p* < 0.001** | ***p* < 0.001** |
| **Daily BM** | | | | | | |
| No/Sometimes | 81 (20.4) | 47.38 (31.46) | 44.21 (32.96) | 74.75 (29.77) | 49.98 (31.21) | 52.84 (30.88) |
| Yes | 316 (79.6) | 53.72 (30.30) | 54.07 (31.27) | 87.87 (19.74) | 61.54 (30.60) | 64.31 (30.66) |
| *Between sub-group comparison* | | *p* = .096 | ***p* = 0.013** | ***p* < 0.001** | ***p* = 0.003** | ***p* = 0.003** |
| **Discomfort doing a BM in a public restroom** | | | | | | |
| Discomfort none of the time (1) | 125 (31.5) | 60.19 (28.97) | 55.42 (32.05) | 87.66 (21.76) | 67.11 (29.34) | 69.66 (28.92) |
| Discomfort a little/some of the time (2) | 190 (47.9) | 52.54 (30.03) | 52.23 (31.14) | 85.16 (22.09) | 57.14 (30.33) | 60.01 (30.13) |
| Discomfort most/all of the time (3) | 82 (20.7) | 40.33 (30.80) | 46.51 (32.69) | 81.52 (25.31) | 51.80 (32.93) | 54.78 (33.94) |
| *Between sub-group comparison overall* | | ***p* < 0.001** | *p* = .143 | *p* = .165 | ***p* = 0.001** | ***p* = 0.002** |
| *1 vs. 2* | | *p* = .068 | | | *p* = 0.014 | *p* = 0.018 |
| *1 vs. 3* | | ***p* < 0.001** | | | ***p* = 0.001** | ***p* = 0.002** |
| *2 vs. 3* | | ***p* = 0.006** | | | *p* = .384 | *p* = .399 |

BM = Bowel movement, CDI = Clostridioides difficile infection, FMT = Faecal microbiota transplantation. Bolded numbers indicate a statistically significant difference. A Bonferroni correction of p < 0.01 for 3 or more comparisons was applied.

* Overall and in all sub-groups, participants were significantly more willing to donate stool to a loved one than for research or new treatments, all *p*'s < 0.01. There were no other differences within sub-groups by location or purpose.

participant's feelings of revulsion or embarrassment, and concerns about donating, collecting, and transporting/delivering stool.

## Discussion

Participants in this study appeared uncertain about donating stool in-centre and at home, with <30% highly willing to donate in either context; a finding which differs markedly from prior research that suggested strong willingness to donate [12]. Of those expressing some willingness to donate, once or twice weekly donations were preferred over one- to three-month periods. Barriers noted in the current study indicate less willingness when the required donation frequency is high or the duration lengthy [8,12,13]. While acknowledging the need to minimise costs associated with donors, this finding suggests that in the longer term it may be advantageous to recruit more donors for less intensive regimes.

Contributors to willingness to donate stool in-centre and at home included the need to understand how donating stool could help patients now or in the future and having positive attitudes towards personally donating stool. In addition, potential in-centre donors who had less need to understand every step of the donation process (suggesting trust or awareness of

**Table 3. Participant's information needs and attitudes by gender, age, and education (N = 397).**

| Characteristic | All | Gender | | Age (in years) | | | | | University educated | |
|---|---|---|---|---|---|---|---|---|---|---|
| | | Male | Female | <21 | 21–30 | 31–40 | 41–50 | >50 | No | Yes |
| **Information needs (1 not at all important to 7 very important)** | | | | | | | | | | |
| Understanding the aim of donating stool. M (SD) | 5.89 (1.42) | 5.78 (1.49) | 6.01 (1.32) | 6.02 (1.31) | 5.90 (1.39) | 5.99 (1.16) | 5.69 (1.75) | 4.92 (2.40) | 5.97 (1.37) | 5.84 (1.44) |
| *Between sub-group comparison* | | *p = 0.112* | | *p = 0.098* | | | | | *p = 0.385* | |
| Understanding how donating stool could help you now or in the future. M (SD) | 5.45 (1.52) | 5.42 (1.49) | 5.49 (1.56) | 5.63 (1.43) | 5.58 (1.47) | 5.20 (1.41) | 5.46 (1.86) | 4.69 (2.10) | 5.73 (1.34) | 5.29 (1.60) |
| *Between sub-group comparison* | | *p = 0.677* | | *p = 0.092* | | | | | ***p = 0.005*** | |
| Understanding how donating stool could help patients now or in the future. M (SD) | 5.96 (1.26) | 5.85 (1.28) | 6.09 (1.22) | 6.13 (0.97) | 6.01 (1.25) | 5.92 (1.15) | 5.92 (1.51) | **4.92 (2.06)[b]** | 6.09 (1.12) | 5.88 (1.33) |
| *Between sub-group comparison* | | *p = 0.054* | | ***p = 0.033*** | | | | | *p = 0.106* | |
| Receiving compensation for donating stool. M (SD) | 4.62 (1.94) | 4.77 (1.99) | 4.45 (1.88) | 4.77 (2.03) | 4.86 (1.75) | 4.50 (1.97) | 4.00 (2.12) | **3.08 (2.14)[c]** | 4.87 (1.99) | 4.46 (1.89) |
| *Between sub-group comparison* | | *p = 0.101* | | ***p = 0.003*** | | | | | ***p = 0.042*** | |
| Out of pocket cost of donating stool, if any. M (SD) | 5.42 (1.64) | 5.31 (1.66) | 5.53 (1.61) | 5.77 (1.51) | 5.48 (1.56) | 5.34 (1.66) | 4.95 (1.95) | 4.77 (1.92) | 5.68 (1.50) | 5.25 (1.70) |
| *Between sub-group comparison* | | *p = 0.170* | | *p = 0.069* | | | | | ***p = 0.011*** | |
| How convenient donating stool would be in terms of logistics. M (SD) | 6.28 (1.14) | 6.23 (1.19) | 6.35 (1.08) | 6.06 (1.13) | 6.30 (1.12) | 6.41 (1.11) | 6.31 (1.24) | 6.08 (1.44) | 6.13 (1.13) | 6.37 (1.14) |
| *Between sub-group comparison* | | *p = 0.283* | | *p = 0.414* | | | | | ***p = 0.041*** | |
| Having to travel in order to donate stool. M (SD) | 6.18 (1.24) | 6.05 (1.36) | 6.32 (1.07) | 6.18 (0.95) | 6.10 (1.32) | 6.22 (1.30) | 6.44 (0.94) | 6.15 (1.52) | 6.23 (1.13) | 6.15 (1.30) |
| *Between sub-group comparison* | | ***p = 0.033*** | | *p = 0.643* | | | | | *p = 0.539* | |
| All information held about you at the stool bank is confidential. M (SD) | 5.93 (1.57) | 5.80 (1.66) | 6.07 (1.45) | 6.27 (1.18) | **5.66 (1.73)[d]** | 6.20 (1.35) | 6.28 (1.28) | **4.92 (2.02)[e]** | 6.08 (1.45) | 5.84 (1.63) |
| *Between sub-group comparison* | | *p = 0.091* | | ***p = 0.001*** | | | | | *p = 0.136* | |
| Understanding every step of the stool donation process before donating. M (SD) | 5.77 (1.45) | 5.69 (1.46) | 5.85 (1.43) | 6.11 (1.22) | 5.64 (1.55) | 5.78 (1.35) | 6.05 (1.34) | **5.08 (1.61)[f]** | 6.01 (1.28) | 5.62 (1.52) |
| *Between sub-group comparison* | | *p = 0.273* | | ***p = 0.049*** | | | | | ***p = 0.009*** | |
| **Attitude–generally towards donating to a stool bank** | | | | | | | | | | |
| Very positive. n (%) | 118 (29.7) | 58 (27.9) | 60 (31.7) | 22 (35.5) | 46 (24.9) | 27 (27.6) | **18 (46.2)[g]** | 5 (38.5) | 49 (32.9) | 69 (27.8) |
| Positive, with reservations. n (%) | 210 (52.9) | 105 (50.5) | 105 (55.6) | 28 (45.2) | **110 (59.5)[h]** | 55 (56.1) | 13 (33.3) | 4 (30.8) | 72 (48.3) | 138 (55.6) |
| Generally negative but realise it is necessary. n (%) | 47 (11.8) | 30 (14.4) | 17 (9.0) | 10 (16.1) | 20 (10.8) | 12 (12.2) | 3 (7.7) | 2 (15.4) | 19 (12.8) | 28 (11.3) |
| Negative. n (%) | 11 (2.8) | **10 (4.8)[a]** | 1 (0.5) | 1 (1.6) | 2 (1.1) | 3 (3.1) | 3 (7.7) | **2 (15.4)[i]** | 4 (2.7) | 7 (2.8) |
| No opinion. n (%) | 11 (2.8) | 5 (2.4) | 6 (3.2) | 1 (1.6) | 7 (3.8) | 1 (1.0) | 2 (5.1) | 0 (0.0) | 5 (3.4) | 6 (2.4) |
| *Between sub-group comparison* | | ***p = 0.037*** | | ***p = 0.014*** | | | | | *p = 0.696* | |
| **Attitude–personally donating stool to a stool bank** | | | | | | | | | | |
| Positive, would generally like to donate. n (%) | 152 (38.3) | 75 (36.1) | 77 (40.7) | 14 (22.6) | 64 (34.6) | **51 (52.0)[j]** | 17 (43.6) | 6 (46.2) | 48 (32.2) | 104 (41.9) |
| Neutral, would depend on the situation. n (%) | 206 (51.9) | 111 (53.4) | 95 (50.3) | **41 (66.1)[k]** | 104 (56.2) | 36 (36.7) | 20 (51.3) | 5 (38.5) | 88 (59.1) | 118 (47.6) |
| Negative, would generally not want to donate. n (%) | 39 (9.8) | 22 (10.6) | 17 (9.0) | 7 (11.3) | 17 (9.2) | 11 (11.2) | 2 (5.1) | 2 (15.4) | 13 (8.7) | 26 (10.5) |

*(Continued)*

**Table 3.** (Continued)

| Characteristic | All | Gender | | Age (in years) | | | | | University educated | |
|---|---|---|---|---|---|---|---|---|---|---|
| | | Male | Female | <21 | 21–30 | 31–40 | 41–50 | >50 | No | Yes |
| *Between sub-group comparison* | | *p* = 0.606 | | *p* = **0.012** | | | | | *p* = 0.084 | |

Bolded numbers indicate a statistically significant difference.

[a] male vs. female, z = 2.6, *p* = 0.009.

[b] >50 vs. < 21 years, *p* = 0.014. >50 vs. 31–40 years, *p* = 0.021.

[c] >50 vs. < 21 years, *p* = 0.031. >50 vs. 31–40 years, *p* = 0.010.

[d] 21–30 vs. 31–40 years, *p* = 0.041.

[e] >50 vs. < 21 years, *p* = 0.033. >50 vs.31-40 years, *p* = 0.039. >50 vs. 41–50 years, *p* = 0.047.

[f] >50 vs. < 21 years, *p* = 0.011.

[g] 41–50 vs. 21–30 (z = 2.7, *p* = 0.008) and 31–40 (z = 2.1, *p* = 0.04) years.

[h] 21–30 vs. < 21 (z = 2.0, *p* = 0.049), 41–50 (z = 3.00, *p* = 0.003), and >50 (z = 2.00, *p* = 0.043) years.

[i] >50 vs. < 21 (z = 2.3, *p* = 0.021), 21–30 (z = 3.5, *p* < 0.001), and 31–40 (z = 2.0, *p* < 0.045) years.

[j] 31–40 vs. < 21 (z = 3.7, *p* < 0.001) and 21–30 (z = 2.8, *p* = 0.005) years.

[k] < 21 vs. 31–40 years (z = 3.6, *p* < 0.001).

the process), and experienced less discomfort doing a BM in a public restroom were more willing to donate. Potential donors who attributed higher importance to knowing how stool donation could personally benefit them, and had less need to understand the aim of donating stool, were more willing to donate stool at home. Addressing potential donor's information needs, minimising discomfort for donors in-centre, and promoting positive attitudes toward personally donating stool appear key to encourage willing donors.

As with prior research [9,11,12], those who were aware of FMT and were blood donors or registered organ donors were more willing to donate stool. However, this association was specific to the donation setting with blood donors more willing to donate stool in-centre, and those aware of FMT and registered organ donors more willing to donate in both contexts. Contrasting with prior research [12], participant characteristics did not significantly contribute to willingness when other factors were considered.

Most participants reported regular BMs, although there were discrepancies in reports of BM frequency [20,23,24], with 58% reporting one BM daily, but only 28% reporting seven BMs weekly (Table 1). This finding suggests that potential donors may be inaccurate in their estimates. Other studies have similarly noted a discrepancy in daily BMs reported depending on the measure used to assess BM frequency [20,23,24]. For example, Lewis and Heaton [23] reported that most participants had a BM once daily. However, when calculations were based on reported weekly frequency, only a third had a daily BM (women 36%, men 38%). Further, when the interval between BMs was considered (range 22 to 27 hours), approximately half on average had a daily BM interval (46.5% women, 50.7% men). Therefore, universal stool banks with preferences for intensive donation regimes should consider asking potential donors to keep a diary of their bowel habits prior to in-person screening.

BMs occurred most often in the morning and took <5 minutes. Two-thirds experienced some discomfort having BMs in public restrooms, and qualitative responses also showed concerns about lack of privacy and hygiene. No gender or age differences in regularity or frequency of BMs emerged [cf. 21–23,25], although females were more likely to have morning BMs, take less time to complete BMs, and experience greater discomfort having BMs in public restrooms [27]. Younger participants were least likely to have morning BMs and most likely to have BMs in the evening or at night. For in-centre collectors, targeting specific demographic

**Table 4. Hierarchical multiple regression analysis of the predictors of willingness to donate stool to a stool bank in-centre and at home and deliver (N = 397).**

| Predictor | Willingness to donate stool in-centre | | | | Willingness to donate stool at home | | | |
| --- | --- | --- | --- | --- | --- | --- | --- | --- |
| | Step 1 | | Step 2 | | Step 1 | | Step 2 | |
| | B (95% CI) | β | B (95% CI) | β | B (95% CI) | β | B (95% CI) | β |
| Gender (0 male, 1 female) | -0.38 (-6.22, 5.45) | -0.01 | -2.32 (-7.14, 2.50) | -0.04 | 4.18 (-2.07, 10.43) | 0.07 | 1.23 (-4.11, 6.58) | 0.02 |
| Age in years | -0.07 (-0.40, 0.25) | -0.02 | -0.15 (-0.42, 0.12) | -0.05 | 0.12 (-0.23, 0.46) | 0.03 | 0.03 (-0.27, 0.32) | 0.01 |
| University education (0 no, 1 yes) | 3.05 (-3.07, 9.16) | 0.05 | 1.58 (-3.52, 6.67) | 0.03 | 3.65 (-2.90, 10.20) | 0.06 | 3.76 (-1.89, 9.41) | 0.06 |
| Blood donor in the past year (0 no/unsure, 1 yes) | 11.10 (2.36, 19.84) | **0.12**\*\* | 5.52 (-1.59, 12.62) | 0.06 | 3.96 (-5.39, 13.32) | 0.04 | -1.73 (-9.60, 6.14) | -0.02 |
| Registered organ donor (0 no/unsure, 1 yes) | 7.28 (0.61, 13.94) | **0.11**\* | 4.76 (-0.73, 10.25) | 0.07 | 7.20 (0.07, 14.34) | **0.11**\* | 5.57 (-0.52, 11.65) | 0.08 |
| Daily BM (0 no/sometimes, 1 yes) | 3.95 (-3.22, 11.13) | 0.05 | -1.71 (-7.63, 4.20) | -0.02 | 8.65 (0.97, 16.33) | **0.11**\* | 2.75 (-3.80, 9.31) | 0.04 |
| Discomfort doing a BM in a public restroom (0 not at all to 5 all the time) | -5.00 (-7.40, -2.60) | **-0.20**\*\*\* | -3.59 (-5.54, -1.63) | **-0.15**\*\*\* | -1.42 (-3.99, 1.15) | -0.06 | 0.14 (-2.03, 2.31) | 0.01 |
| Heard of FMT prior to study (0 no, 1 yes) | 7.97 (1.90, 14.04) | **0.13**\*\* | 2.92 (-2.21, 8.05) | 0.05 | 7.33 (0.83, 13.83) | **0.12**\* | 3.37 (-2.32, 9.06) | 0.05 |
| Understanding the aim of donating stool | | | -2.10 (-4.67, 0.47) | -0.10 | | | -4.26 (-7.11, -1.41) | **-0.19**\*\* |
| Understanding how donating stool could help you now or in the future | | | 1.08 (-0.96, 3.12) | 0.05 | | | 2.28 (0.02, 4.55) | **0.11**\* |
| Understanding how donating stool could help patients now or in the future | | | 3.58 (0.51, 6.66) | **0.15**\* | | | 6.10 (2.69, 9.50) | **0.24**\*\*\* |
| Receiving compensation for donating stool | | | 0.90 (-0.58, 2.37) | 0.06 | | | -0.07 (-1.70, 1.57) | -0.00 |
| Out of pocket cost of donating stool, if any | | | 0.76 (-1.05, 2.56) | 0.04 | | | 0.29 (1.72, 2.29) | 0.02 |
| How convenient donating stool would be in terms of logistics | | | 0.01 (-2.60, 2.62) | 0.00 | | | -1.27 (-4.16, 1.63) | -0.05 |
| Having to travel in order to donate stool | | | 0.31 (-1.91, 2.53) | 0.01 | | | 2.23 (-0.24, 4.69) | 0.09 |
| All information held about you at the stool bank is confidential | | | -0.50 (-2.25, 1.25) | -0.03 | | | -1.71 (-3.66, 0.23) | -0.08 |
| Understanding every step of the stool donation process before donating | | | -2.68 (-4.73, -0.62) | **-0.13**\*\* | | | -0.16 (-2.43, 2.12) | -0.01 |
| Attitude towards personally donating (0 neutral/negative, 1 positive) | | | 35.45 (30.27, 40.63) | **0.56**\*\*\* | | | 32.62 (26.88, 38.56) | **0.50**\*\*\* |
| $\Delta R^2$ | 0.125 | | 0.322 | | 0.074 | | 0.298 | |
| F change | $F(8, 388) = 6.92$, $p < 0.001$ | | $F(10, 378) = 2.02$, $p < 0.001$ | | $F(8, 387) = 3.88$, $p < 0.001$ | | $F(10, 377) = 17.93$, $p < 0.001$ | |

\*\*\* $p < 0.001$.

\*\* $p < 0.01$.

\* $p < 0.05$. Bolded numbers indicate a significant predictor.

BM = Bowel movement.

Willingness to donate stool in-centre, $F(18, 378) = 16.98$, $p < 0.001$. Willingness to donate stool at home, $F(18, 377) = 12.44$, $p < 0.001$.

groups to donate at non-peak times (e.g., women early in the morning; younger ages in the evening) may be feasible. Potential donors should also be screened early on in recruitment regarding their discomfort having BMs in public facilities. Recruitment materials should emphasise privacy and hygiene measures implemented in collection facilities to reassure potential donors.

**Table 5. Themes identified in qualitative analysis of responses to motives (N = 391) and barriers for stool donation (N = 390).**

| Themes and sub-themes (n, % responses) | Exemplar quotes |
|---|---|
| *Motives (n = 483 responses)* | |
| Altruism (n = 295, 61%)<br>• Impure altruism (89%)<br>• Egalitarian warm glow (5%)<br>• Reciprocal altruism (3%)<br>• Reluctant altruism (3%) | "It would be a great feeling knowing that something I did saved someone's life. Much like how blood donations save lives, so do stool donations."<br>"Feeling like I am able to help someone with a condition that may be serious and or life-threatening. Feeling as though I can "make a difference" and improve the quality of someone else's life."<br>"I donate blood and plasma regularly. It is my way of helping the world how I can. I can't change big things, but things like this, is within my control. I can't help every suffering person, but I can make a difference to one. Same with this."<br>"Giving back to the community/leaving a positive influence on society."<br>"Who knows what health conditions I might have in the future and need medical help. Where I'm healthy I hope I can help. The biggest motivation I have to help would be if a family member has been afflicted with this disease or a very close friend."<br>"To help someone in need knowing that I may be able to save a life if there is a shortage of people willing to donate." |
| Compensation (n = 65, 13.5%) | "Financial. Sorry but the combination of embarrassment and time spent being healthy and delivery of the stool has to make up for lost time."<br>"To relieve the suffering of others and give back to the community. As a secondary reason, I would also be motivated by compensation, such as drinks and snacks offered to me after the donation."<br>"Financial reasons would certainly enhance the likelihood of donation but if there was a sound enough reason—helping others—and an easy anonymous process that would help." |
| Benefits to recipients outweigh the cost of donating (n = 44, 9%) | "Because it's essentially a non-invasive procedure that doesn't cost me anything so why wouldn't I want to help someone if I can?"<br>"To be honest, other than logistics (such as location) I cannot find many reasons not to. If my stool, that otherwise would go to waste could potentially have positive health benefits for another person, or contribute to research to ensure well-being of others, then that would be enough of a reason for me." |
| Helping science (n = 33, 7%) | "I would be donating to assist in treating their key health issues and hope that my sample would also be used in developing alternate treatments (research)."<br>"I believe that participating in research is very important, to help other people and also to advance research. Also hopefully reduces stigma around topics such as this one, normalises the concept to those around me." |
| Greater awareness of CDI and FMT (n = 18, 4%) | "Help others. Success rate is important i.e. we know organ donation and blood donation saves lives but does this. More emphasis on success rate is important."<br>"Compensation would be motivating, but also the understanding of just HOW helpful it could potentially be." |
| Donation is convenient and logistically feasible (n = 17, 3.5%) | "Easy to do the donation within the acceptable range of transportation."<br>"If it were easy to do, I'd decide to do it because it would help someone in need and depending on the circumstances could potentially save a life. Receiving compensation would be another big motivator. But as long as the process was easy and preferably fairly quick to fulfil, I'd be quite motivated to do it as I am a lazy/impatient person." |
| Personal involvement (n = 11, 2%) | "I believe it would help a lot to have knowledge of this disease and to have some kind of experience with it (e.g., a family member or friend) as this usually give more insight into the issues and makes people more willing to donate." |

(*Continued*)

**Table 5.** (Continued)

| Themes and sub-themes (n, % responses) | Exemplar quotes |
|---|---|
| *Barriers (n = 481 responses)* | |
| Logistics (n = 192, 39.9%)<br>• Having to travel to donate (47%)<br>• General inconvenience (15%)<br>• Lack of flexibility in scheduling or donation requirements (28%)<br>• Lengthy donation process (8%)<br>• Out of pocket costs (2%) | "The high frequency required (compared to blood donation which is once every three months) and the location of a donation centre (if it wasn't convenient)."<br>"If I had to do it too often, making it a chore, or if the method of donation were inefficient and a bit of a commute. Also, if I had to pay the express postage myself, I think I would be much less likely to continue with my service."<br>"If I had to provide the stool at the stool bank itself instead of doing so from home and then delivering it. Or if the requirements for donating were too strict it would become difficult, e.g. 6–7 times a week would be too hard or maybe tiring to have to deliver each time."<br>"If I cannot batch the stool delivery with other activities I do in the day and have to specifically go out to do it." |
| Concerns about capabilities (n = 83, 17.3%)<br>• Difficult/complicated process (51%)<br>• Co-ordinating time of BMs (15%)<br>• Health/quality of sample (28%)<br>• Fear of needles/blood tests (6%) | "The process and whether it is an easy process to donate and send it off."<br>"Long complicated or too frequent communications. Long wait times or complicated way of delivering the sample."<br>"It's not something I can time or know when it's going to happen as such so it isn't like a doctor's appointment or blood donation where you know when you turn up it is going to be able to take place."<br>"Giving blood. I understand that's part of the process but needles are a huge no no for me." |
| Disgust (n = 80, 16.6%)<br>• Disgust at the thought of donating stool (14%)<br>• Disgust at having to collect stool (32.5%)<br>• Unhygienic conditions (38%)<br>• Disgust at transporting/delivering stool (16%) | "The idea of putting my stool in another humans body—no matter the benefit, it's hard to digest."<br>"If the process involves me having to poop into a container and I accidentally touch my poooo:(((" <br>"Public-type restroom or otherwise uncomfortable facilities."<br>"The process of getting it to the stool bank—where is it?? If it's in/near the hospital here I have to drive in, pay for parking, and carry crap through the busy streets. Imagine people without cars doing this too. It's just very weird. I'd like to know more about the containers. The lid coming off in a backpack would be a disaster." |
| Discomfort (n = 78, 16.2%)<br>• Lack of anonymity or privacy (47%)<br>• Embarrassment (40%)<br>• Discomfort doing a BM in a public facility (13%) | "If I would meet face to face the people who would examine my stool."<br>"Everybody knowing what I'm doing. Discreet way would be better to manage."<br>"Embarrassment. Friends or family if they were to know about it being embarrassed of me, or I of myself during any part of the process."<br>"The handling of the stool and having to discuss the process with others who may not understand/judge the reason for the donation."<br>"I would not want to donate stool I produced in a public bathroom, even one at the stool bank itself. I would far prefer being able to poop at home and bring it in that way." |
| Availability (n = 41; 8.5%) | "The difficulties are largely practical, to do with how to access the donation places and manage to do it without major disruptions to my work schedule."<br>"Not being able to fit it in around work and picking up the kids from school." |
| Lack of compensation (n = 7; 1.5%) | "If it was hard to donate and I had to go out of my way to do it for no reward." |

CDI = Clostridioides difficile infection.

Prominent information needs focused on convenience of stool donation and travel required in order to donate. Information needs also differed by gender, age, and education. For instance, convenience was of greater concern for university educated participants, and travel requirements were more important for females. These information needs align with

logistics (e.g., travel, donation requirements, flexibility) and concerns about capabilities to donate as the main barriers identified to stool donation by our sample, and also concur with prior research [12,14–16]. Targeting donor recruitment efforts in locations close to stool banks, clinics or collection facilities would minimise inconvenience and improve logistics. Tying stool donation to another donation type (e.g., blood donation) [43], or another activity that is an established part of a donor's routine (e.g., workplace donor programs) may also decrease barriers. Potential donors should be provided with a 'donor journey map' at an early stage to build their confidence, and provide reassurance that donors have limited interaction with their donation and staff in-centre, once they have donated.

Altruism, most often helping others generally or those specifically with CDI, was identified as the main reason participants would consider donating. This theme is consistent with prior research [9,12], as is the finding in this study that understanding how stool donation could help patients was a significant positive predictor of willingness to donate. Financial compensation was a motivator of lesser importance, and something that would reinforce participant's primary helping motivation. Donor recruitment materials should emphasise how stool donation helps patients and incorporate messages highlighting the large benefit to recipients versus small cost to donors ratio and helping to advance science.

Strengths of this study include the perspective of potential donors in Australia where stool banks and the need for community donors are rapidly developing [44,45]. It provides information not currently available on potential donors' bowel habits, discomfort donating outside the home environment, and information needs. It extends research on engaging people in this form of donation which compared to blood and organ donation is relatively novel, with little information available. This study is limited by the potential for participants who were already interested and motivated to donate stool to have self-selected to participate. Further, information about participant's occupation or location was not collected, preventing a consideration of the influence of these characteristics on willingness to donate. It is possible that participants in this study appeared less willing to donate than the previous Canadian study [12] due to this latter study including participants who were university-affiliated health professionals and thus potentially more willing to donate. Moreover, there is the potential for differences in willingness to donate stool based on region (e.g., urban vs. rural) or country-specific delivery of healthcare systems (e.g., public and private, non-profit and for-profit) and the perceived or actual costs for donors. Country-specific differences in protocols (e.g., age, facilities) and changes in terminology and processes due to advances in automated facilities (e.g., washed microbiota transplantation vs. manual faecal microbiota transplantation [19]) may also impact willingness and acceptability of stool donation. This study is cross-sectional, and therefore limited by its focus on willingness to donate, rather than donor behaviour. Although study eligibility criteria mirrors that used broadly for community donors to stool banks, we relied on self-report rather than objective measures. Finally, although the contributors examined in this study explained a significant proportion of variance in willingness to donate, the variance left unexplained suggests other unidentified factors may inform potential donors' willingness and these require further exploration.

In conclusion, this study revealed that most Australian community members are uncertain about donating stool. This study identified the importance of and need to consider bowel habits, level of comfort donating in-centre, and potential donor's information needs as early as possible in the recruitment and screening process to optimise their willingness to donate. Results also confirm that in order to increase the numbers of willing donors, it is critical for universal stool banks and FMT programs to examine logistics associated with recruitment and donation processes and implement strategies that increase flexibility, feasibility, and

comprehensibility for potential donors who are constrained by time, knowledge deficits, and limited access to stool banks and facilities.

## Supporting information

**S1 Appendix. Pre-screening and main surveys.**
(DOCX)

**S2 Appendix. Information presented to participants about CDI, FMT, and stool donation.**
(DOCX)

## Acknowledgments

We acknowledge Abby Edwards for assistance with data collection and coding. Australian governments fund Australian Red Cross Lifeblood for the provision of blood, blood products, and services to the Australian community.

## Author Contributions

**Conceptualization:** Melissa K. Hyde, Barbara M. Masser.

**Data curation:** Melissa K. Hyde, Barbara M. Masser.

**Formal analysis:** Melissa K. Hyde, Barbara M. Masser.

**Funding acquisition:** Melissa K. Hyde, Barbara M. Masser.

**Investigation:** Melissa K. Hyde, Barbara M. Masser.

**Methodology:** Melissa K. Hyde, Barbara M. Masser.

**Project administration:** Melissa K. Hyde, Barbara M. Masser.

**Writing – original draft:** Melissa K. Hyde, Barbara M. Masser.

**Writing – review & editing:** Melissa K. Hyde, Barbara M. Masser.

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
