## [Decision Letter · Decision Letter 0]

5 Oct 2020

PONE-D-20-21058

Determinants of community member’s willingness to donate stool for faecal microbiota transplantation

PLOS ONE

Dear Dr. Hyde,

Thank you for submitting your manuscript to PLOS ONE. After careful consideration, we feel that it has merit but does not fully meet PLOS ONE’s publication criteria as it currently stands. Therefore, we invite you to submit a revised version of the manuscript that addresses the points raised during the review process.

I agree with the reviewers concerning this manuscript. They each raise relatively minor points and those points are aimed at clarifying the design and analysis of the data. As such, it should be relatively straightforward to address their concerns.

I look forward to a revision.

We look forward to receiving your revised manuscript.

Kind regards,

Rick K. Wilson, Ph.D.

Academic Editor

PLOS ONE

Journal Requirements:

"This work was funded in part by an award made from The University of Queensland (UQ) Vice-Chancellor’s Strategic Funds (BM) and internal funds granted by the UQ School of Psychology (MH). The funders did not play any role in the study design, data collection and analysis, decision to publish, or preparation of the manuscript.".

i) Please provide an amended statement that declares *all* the funding or sources of support (whether external or internal to your organization) received during this study, as detailed online in our guide for authors at http://journals.plos.org/plosone/s/submit-now.  Please also include the statement “There was no additional external funding received for this study.” in your updated Funding Statement.

ii) Please include your amended Funding Statement within your cover letter. We will change the online submission form on your behalf.

Reviewers' comments:

Reviewer's Responses to Questions

**Comments to the Author**

1. Is the manuscript technically sound, and do the data support the conclusions?

Reviewer #1: Yes

Reviewer #2: Yes

2. Has the statistical analysis been performed appropriately and rigorously? 

Reviewer #1: Yes

Reviewer #2: Yes

3. Have the authors made all data underlying the findings in their manuscript fully available?

Reviewer #1: Yes

Reviewer #2: Yes

4. Is the manuscript presented in an intelligible fashion and written in standard English?

Reviewer #1: Yes

Reviewer #2: Yes

5. Review Comments to the Author

Reviewer #1: Hyde and Masser present a large survey of factors influencing participation as a stool donor for FMT from Australia. They explore participant demographics, bowel habit, factors that may attract or deter donors, and motivators to donation, exploring these quantitatively and qualitatively. They discuss how knowledge of these push and pull factors can be applied in clinical practice by stool banks to influence their policies regarding donor recruitment and retention.

I have very little concerns with the study design, the analysis, the presentation of data and the interpretation – these are all great. There are just a few smaller factors that I think are also worthy of consideration/ amendment:

-Title – I think should be members’ rather than member’s.

-Consistency required with spelling, e.g. ‘faecal’ in title but ‘fecal’ in abstract.

-Could you explain a little more about how people were recruited to take the survey? It is important to have this information to give more insight into the population completing it, and any potential biases in participant recruitment.

-Was any information recorded about work of participants? It would be interesting to see if a connection to healthcare influenced outlook about this.

-I am not sure that any data is provided regarding where in Australia that participants came from. It might at least be interesting to know if there was any difference in outlook between people in smaller/ more rural communities from larger/ urban connurbations? Where differences exist between outcome in this study compared to previous studies from other countries/ regions – could the investigators propose any country-specific regions why these might differ, e.g. aspects related to the delivery of healthcare systems?

Reviewer #2: This research titled “Determinants of community member’s willingness to donate stool for fecal microbiota transplantation” has the merit to represent the first study of the Australian community’s willingness to donate stool. This study offers some good suggestions for stool banks to increase the numbers of willing donors. Comments:

1.This is an interesting and practical survey on potential donors’ attitudes and willingness to FMT. This mainly reflected the situation in Australia. However, the methods and protocol on donor screening in the global varies a lot in different areas or countries. I suggest authors to discuss and the latest consensus report from the FMT-standardization study group. The recommended donor population is younger than the mean age of investigated population in the current study. Importantly, the location for collecting stool is only in the specific room within the FMT center. This is the biosafety requirement according to the consensus. At least this is very important for FMT center. I suggest discuss the direction and different choices under the different regulations. Readers of the journal should know the general view on the donor screening and related laboratory process.

Fecal Microbiota Transplantation-standardization Study Group. Nanjing consensus on methodology of washed microbiota transplantation [published online ahead of print, 2020 Jul 21]. Chin Med J (Engl). 2020;10.1097/CM9.0000000000000954. doi:10.1097/CM9.0000000000000954

2.Line 116 - Information presented to participants about CDI and FMT was particularly important. I would kindly ask the authors to provide the detailed short paragraphs describing CDI and FMT (even as Supplementary material).

3.Question about Table 1 - The data of BMs daily frequency <1 and BMs weekly frequency <7 are inconsistent, which you explained in the discussion section may be due to inaccurate assessment of participants. This raises questions about the reliability of the questionnaire data.

4.Line 178 - Can the author describe the scoring rules for willingness to donate stool in detail?

5. According to the latest report, FMT has been named washed microbiota transplantation (WMT) based on the latest technology. Participants may make different choices after learning about these new developments. You can add these to the discussion section.

6. PLOS authors have the option to publish the peer review history of their article (what does this mean?). If published, this will include your full peer review and any attached files.

Reviewer #1: No

Reviewer #2: No

---

## [Author Response · Author response to Decision Letter 0]

11 Nov 2020

Dear Professor Wilson and Reviewers, 

Thank you very much for the feedback and for the opportunity to undertake revisions. We appreciate the useful suggestions from the reviewers and have endeavoured to incorporate these where possible. Our responses to reviewer feedback are below. Changes to the manuscript are identified using track changes. 

Please note as per our cover letter, since our original PLOS ONE submission we subsequently published an article in Transfusion that used a subset of participants to explore a wholly different focus on the role of ambivalence in decisions about stool donation and used an independent set of variables. We have noted the link to the Transfusion paper in the text of our revised PLOS ONE manuscript on p. 6. 

We have separately addressed in our cover letter the additional journal requirements outlined in the manuscript decision letter.

Thank you once again for considering our manuscript and we look forward to hearing from you.

Yours sincerely,

The Authors

REVIEWER 1

Feedback: I have very little concerns with the study design, the analysis, the presentation of data and the interpretation – these are all great. 

Response: Thank you very much for your kind evaluation of our work. 

Feedback: There are just a few smaller factors that I think are also worthy of consideration/ amendment: 

a) Title – I think should be members’ rather than member’s.

Response: a) Thank you, the title has now been corrected.

Feedback: b) Consistency required with spelling, e.g. ‘faecal’ in title but ‘fecal’ in abstract.

Response: b) Our apologies for the error and thank you for noting. We have now ensured all spelling is consistent.

Feedback: c) Could you explain a little more about how people were recruited to take the survey? It is important to have this information to give more insight into the population completing it, and any potential biases in participant recruitment.

Response: c) Thank you for this suggestion. We have added further information in the Method and Materials section on p. 5 and p. 6 about how we recruited participants to do the survey. Specifically, we have noted the approximate number of Australians who were active on the system, the proportion of men, and that participants self-selected into the study and completed a pre-screening survey to confirm their eligibility before being invited to complete the main survey. Pre-screening questions have also been included as Supplementary Information (S1 Appendix).

Feedback: d) Was any information recorded about work of participants? It would be interesting to see if a connection to healthcare influenced outlook about this.

Response: d) We agree this information would be interesting to have. Unfortunately, we did not ask participants about their occupation and we have now noted this as a limitation on p. 35.

Feedback: e) I am not sure that any data is provided regarding where in Australia that participants came from. It might at least be interesting to know if there was any difference in outlook between people in smaller/ more rural communities from larger/urban connurbations? Where differences exist between outcome in this study compared to previous studies from other countries/ regions – could the investigators propose any country-specific regions why these might differ, e.g. aspects related to the delivery of healthcare systems?

Response: e) Thank you for this suggestion. We agree it would be interesting to explore this aspect. Unfortunately, we did not collect data about location, however we have included this as a limitation based on the potential for differences across region or countries on p. 35. 

REVIEWER 2

Feedback: This is an interesting and practical survey on potential donors’ attitudes and willingness to FMT.

Response: Thank you, we appreciate your kind evaluation of our work.

Feedback: 1) This mainly reflected the situation in Australia. However, the methods and protocol on donor screening in the global varies a lot in different areas or countries. I suggest authors to discuss and the latest consensus report from the FMT-standardization study group. The recommended donor population is younger than the mean age of investigated population in the current study. Importantly, the location for collecting stool is only in the specific room within the FMT center. This is the biosafety requirement according to the consensus. At least this is very important for FMT center. I suggest discuss the direction and different choices under the different regulations. Readers of the journal should know the general view on the donor screening and related laboratory process.

Response: 1) Thank you for raising this important point. We have now included discussion that screening criteria and protocols vary globally and have included an example of this with references to protocols including the suggested Nanjing consensus from the FMT-standardization Study Group on p. 5-6. 

We have also included information that there is variability in stool collection procedures so that readers are aware and noted that this variability in process may impact willingness and acceptability of donation (p. 4 and p. 35)

Feedback: 2) Line 116 - Information presented to participants about CDI and FMT was particularly important. I would kindly ask the authors to provide the detailed short paragraphs describing CDI and FMT (even as Supplementary material).

Response: 2) On the Reviewer’s advice, we have now provided the detailed short paragraphs about CDI and FMT as Supplementary Information (S2 Appendix).

Feedback: 3) Question about Table 1 - The data of BMs daily frequency <1 and BMs weekly frequency <7 are inconsistent, which you explained in the discussion section may be due to inaccurate assessment of participants. This raises questions about the reliability of the questionnaire data.

Response: 3) Thank you for raising this point. We have now further explained that this inaccuracy in estimating weekly BMs is not unique to our study and has occurred and been noted in prior research (p. 33). It was our knowledge of this research that prompted us to include multiple assessments so readers could obtain a fuller picture about the bowel habits of participants.

Feedback: 4) Line 178 - Can the author describe the scoring rules for willingness to donate stool in detail?

Response: 4) We have included further detail about the scoring of willingness on p. 7.

Feedback: 5) According to the latest report, FMT has been named washed microbiota transplantation (WMT) based on the latest technology. Participants may make different choices after learning about these new developments. You can add these to the discussion section.

Response: 5) 5. Thank you for this suggestion. We have now noted the following in the Discussion on p. 35: 

“Country-specific differences in protocols (e.g., age, facilities) and changes in terminology and processes due to advances in automated facilities (e.g., washed microbiota transplantation vs. manual faecal microbiota transplantation [19]) may also impact willingness and acceptability of stool donation.”

---

## [Editor Report · Decision Letter 1]

26 Nov 2020

Determinants of community members' willingness to donate stool for faecal microbiota transplantation

PONE-D-20-21058R1

Dear Dr. Hyde,

We’re pleased to inform you that your manuscript has been judged scientifically suitable for publication and will be formally accepted for publication once it meets all outstanding technical requirements.

I want to thank you for being so responsive to the reviewers. I think that the manuscript now reads much better and clarifies minor concerns held by the reviewers. I see no reason to ask the reviewers to comment further on the revision.

Kind regards,

Rick K. Wilson, Ph.D.

Academic Editor

PLOS ONE
---

## [Editor Report · Acceptance letter]

2 Dec 2020

PONE-D-20-21058R1 

Determinants of community members’ willingness to donate stool for faecal microbiota transplantation 

Dear Dr. Hyde:

I'm pleased to inform you that your manuscript has been deemed suitable for publication in PLOS ONE. Congratulations! Your manuscript is now with our production department. 

Kind regards, 

on behalf of

Dr. Rick K. Wilson 

Academic Editor

PLOS ONE